# VerMCTS: Synthesizing Multi-Step Programs using a Verifier, a Large Language Model, and Tree Search

**David Brandfonbrener**[*]  **Simon Henniger**[†]  **Sibi Raja**[‡]  **Tarun Prasad**[‡]

**Chloe Loughridge**[‡]  **Federico Cassano**[§]  **Sabrina Ruixin Hu**[‡]  **Jianang Yang**[¶]

**William E. Byrd**[‖]  **Robert Zinkov**[††]  **Nada Amin**[‡]

## Abstract

Large Language Models (LLMs) can generate useful code, but often the code they generate cannot be trusted to be sound. In this paper, we present VerMCTS, an approach to begin to resolve this issue by generating verified programs in Dafny and Coq. VerMCTS uses a logical verifier in concert with an LLM to guide a modified Monte Carlo Tree Search (MCTS). This approach leverages the verifier to gain intermediate feedback inside the search algorithm by checking partial programs at each step to estimate an upper bound on the value function. To measure the performance of VerMCTS, we develop a new suite of multi-step verified programming problems in Dafny and Coq. In terms of pass@$T$, a new metric which computes the pass rate given a budget of $T$ tokens sampled from the LLM, VerMCTS leads to more than a 30% absolute increase in average pass@5000 across the suite over repeated sampling from the base language model.

## 1   Introduction

Large Language Models (LLMs) are increasingly used for generating code, but the code needs to be inspected and possibly re-generated if it doesn't satisfy the user [Zhong and Wang, 2023]. We propose to partially shift the burden of checking code, from the user to the LLM, by generating code in a verification-aware programming language like Dafny or Coq, prompting for specifications and proofs of correctness in addition to code that can then be formally verified. In such a system, the user can focus their attention on the specifications, and less on the code and proofs with the assurance that the generated output has passed the verifier. Our approach couples imprecise generative reasoning from an LLM with logical reasoning from a program verifier. The LLM contributes fruitful suggestions and the verifier ensures soundness.

As a motivating example, consider this prompt: *In Dafny, write an ADT for arithmetic expressions comprising constants, variables, and binary additions. Then write an evaluator taking an expression and an environment (a function that takes a variable name and returns a number) and returning the number resulting from evaluation. Then write an optimizer taking an expression and returning an expression with all additions by 0 removed. Then prove that the optimizer preserves the semantics as defined by the evaluation function.*

To aid a language model to tackle this task, we introduce VerMCTS, an algorithm that combines a verifier and tree search with a language model to synthesize verified programs. An overview of the

---

[*]Kempner Institute at Harvard University, [‡] Harvard University, [†] TU München, [§] Northeastern University, [¶] Million.js, [‖] University of Alabama at Birmingham, [††] University of Oxford
Correspondence to `namin@seas.harvard.edu`

MATH-AI Workshop at the 38th Conference on Neural Information Processing Systems (NeurIPS 2024).

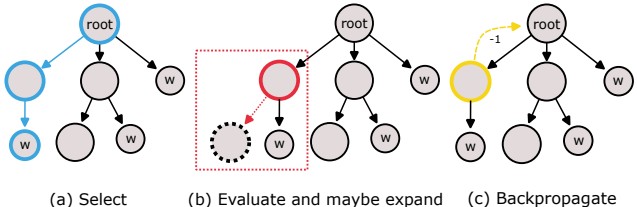

(a) Select     (b) Evaluate and maybe expand     (c) Backpropagate

Figure 1: Overview of VerMCTS. The search tree is visualized with "widen" nodes marked with $w$. (a) A leaf node is selected as in standard MCTS. (b) The selected node is evaluated and maybe expanded. If the selected node is a widen node, then it's parent is selected and maybe expanded (i.e. made wider). See Figure 2 for a detailed description. (c) Once we have a value from the evaluate and maybe expand algorithm, we backpropagate that value up the tree. This figure illustrates the special case where we observed a failure, so no node is added and the score is -1.

algorithm is described in Figure 1 and Figure 2 and the details are presented in Section 2. VerMCTS creates a search tree with progressive widening so it is capable of handling large action spaces defined by lines of code. Within this search tree both expansion and evaluation are guided by the verifier which acts as a computationally cheap (relative to the LLM) upper bound on the value function in the code synthesis MDP, as we show in Section 2.

To evaluate VerMCTS we introduce a suite of 15 challenge problems (9 in Dafny and 6 in Coq). This suite probes essential skills needed for general verified programming like constructing algebraic data types, defining functions, and writing inductive proofs.

On this suite of problems we compare VerMCTS with several baselines including repeated sampling of full programs from the base model, an advanced prompting technique that uses access to the error messages generated by the verifier called Reflexion [Shinn et al., 2023], and a traditional version of MCTS. We quantify performance in terms of pass@$T$, the pass rate within a budget of $T$ tokens. VerMCTS outperforms the baselines substantially, leading to a 30% absolute average performance improvement over repeated sampling from the base model. Note this repeated sampling is a strong baseline, similar to a pass@$k$ evaluation often used as a skyline in program generation. Moreover, for several problems VerMCTS solves problems that are not solved at all by other methods within the given budget.

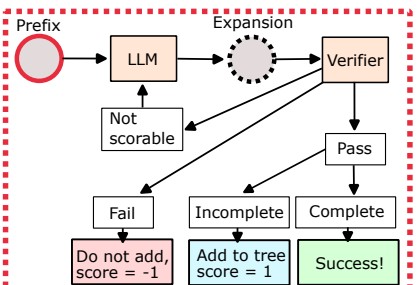

Figure 2: Evaluate and maybe expand. Given a prefix, we query the LLM for expansions until the verifier is able to return a score. If that score is a failure, we do not add the node to the tree, but update the parent with a value of -1. If the score is pass, then we check if the program is complete. If incomplete, we add the expansion to the tree with a score of 1. If complete, we have found a successful program to return.

## 2 Method: VerMCTS

Our main contribution is to define a search algorithm inspired by MCTS that leverages a verifier and LLM to search for verified programs. We call this method *VerMCTS*. In this section, we first present the Markov Decision Process that we consider as the environment for verified program synthesis and then present VerMCTS in detail. VerMCTS is a variant of traditional MCTS that incorporates the LLM as a prior to generate candidates and the verifier as a heuristic to evaluate partial programs.

### 2.1 MDP for verified program synthesis

We formulate our multi-step verified synthesis problem as a Markov Decision Process (MDP) $\mathcal{M} \coloneqq (\mathcal{S}, \mathcal{A}, T, r, H)$ defined by the LLM and the verifier. Here, $\mathcal{S}$ refers to the state space, $\mathcal{A}$ refers to the action space, $T \colon \mathcal{S} \times \mathcal{A} \to \mathcal{S}$ refers to the (deterministic) transition dynamics of the environment, $r \colon \mathcal{S} \to \mathbb{R}$ refers to the reward function, and $H$ is the finite horizon (i.e. a limit on the number of transitions). Defining the MDP just consists of defining these four objects. The state, action, transition dynamics, and reward are defined as follows:

- Each state $s \in \mathcal{S}$ is a string consisting of the initial user prompt and a partial program.

- Actions $a \in \mathcal{A}$ are strings that represent a unit of a program. In Dafny each line is an action. In Coq each "command" (ending with a dot '.') is an action. We also limit the number of tokens in an action.

- The transition dynamics are just defined by string concatentation: $T(s, a) = s + a$.

- The reward function $r$ is defined by the verifier for a given verified programming language and is only defined on complete programs. This terminal reward is 1 if the complete program is accepted and -1 if it is rejected. The reward is 0 for incomplete programs.

With this simple MDP in place, we can define our search algorithm for finding verified programs.

## 2.2 VerMCTS

Given this MDP with finite actions and deterministic dynamics, it would be possible to run standard MCTS to learn a stochastic policy, but the action space is much too large for this to be practical. Instead, we build a search algorithm inspired by MCTS that can leverage the LLM as a prior for program synthesis and the verifier to evaluate partial programs. Both components are key for a successful search in this large space.

Standard MCTS consists of four steps: select, expand, evaluate, and backpropagate. Our algorithm leaves the select and backpropagate steps essentially unchanged. We modify and combine the expand and evaluate steps to leverage the power of the LLM and the verifier in tandem. Our full algorithm is illustrated in Figure 1. In this section we first discuss progressive widening and then go through each step of VerMCTS in turn.

**Progressive widening.** To allow for potentially infinite width while still efficiently conducting deep searches, we adapt an idea from classical work on MCTS to progressivly widen nodes in the tree [Chaslot et al., 2008, Couëtoux et al., 2011]. In that work, the number of children available at a given node scales explicitly with the number of visits. In our setting since adding a child node requires an expensive call to the LLM, we instead opt to add a

---

**Algorithm 1** Evaluate and (maybe) expand

1: **Input:** string $s$, depth limit $L$
    LLM: string $\rightarrow$ completion
    Verifier: string $\rightarrow \{-1, 0, +1\}$
2: **Output:** value $v(s)$, (optional) child node
3: `score` $\leftarrow 0$
4: `depth` $\leftarrow 0$
5: $a \leftarrow$ `""`
6: **while** `score` $= 0$ and `depth` $< L$ **do**
7:     $a \leftarrow a +$ LLM$(s + a)$
8:     `score` $\leftarrow$ Verifier $(s + a)$
9:     `depth` $\leftarrow$ `depth` $+ 1$
10: **end while**
11: **if** `score` $= -1$ or `depth` $= L$ **then**
12:     **return** $-1$, None
13: **else**
14:     **return** $+1$, $s + a$
15: **end if**

---

"widen" child to each node that is assigned 0 value and can be selected via the selection procedure described below. This allows the scoring mechanism to prioritize when to expand a node by essentially setting a prior that unexplored branches have 0 value. When the widen node $w$ with parent $s$ is selected, instead of adding a child to $w$, we add a child to $s$ (i.e. add a sibling to $w$). In this way, the tree can grow wider over the search process.

**Selection: priors and UCT.** We use a standard MCTS selection step, but we set a prior for the UCT (upper confidence bound for trees) bonus as in PUCT [Rosin, 2011, Silver et al., 2016]. We choose to let the prior $p = 1.0$ for standard nodes and let $p = p_{widen} < 1.0$ for widen nodes be a hyperparameter that we tune. This basic heuristic gives the model a preference to select the standard nodes which encourages deeper search trees while still allowing for potentially infinite width if needed. With this choice, the score of a node $s$ is:

$$\text{score}(s) = p_s \cdot c_{UCT} \sqrt{\frac{\log N_{parent}}{N_s}} + \frac{\sum_{i=1}^{N} v_i}{N_s} \tag{1}$$

where $p_s$ is the prior at this node, $c_{UCT}$ is a global exploration coefficient, $N_{parent}$ is the number of visits at the parent node, $N_s$ is the number of visits at this node, and $v_i$ is the estimated value at the $i$th visit to $s$. Note that this selection procedure has two hyperparameters: $c_{UCT}$ and $p_{widen}$ that encourage selecting more rarely visited nodes and widen nodes respectively.

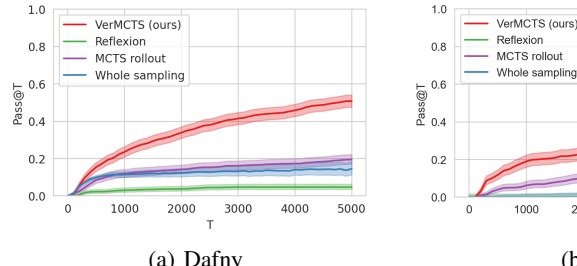

(a) Dafny                    (b) Coq

Figure 3: Average results for pass@T vs. T (the number of tokens) for various baseline methods on our suite of programming problems in Dafny and Coq.

**Combining expansion and evaluation.**  Traditionally, MCTS will first expand a node into children and evaluate it either by simulated rollouts [Chaslot et al., 2008, Zhang et al., 2023a] or a learned value function [Silver et al., 2016]. Neither of these methods is a good fit for our problem because generating rollouts requires many expensive calls to the LLM and learning a value requires large amounts of training data. Moreover, both methods give noisy signal, but in our setting we have access to the ground truth verifier.

Beyond being noiseless, the verifier has one more important property: if a partial program fails the verifier, no subsequent completion can yield success. So, we want to make sure that we never add to the tree any expansion that is a guaranteed failure. Doing this require explicitly linking expansion to evaluation where we evaluate the node and *maybe* expand it, as formalized in Algorithm 1.

In addition to only adding nodes with potential to the tree, we want to leverage the verifier to cheaply evaluate partial programs without extra calls to the LLM. Explicitly, from a node containing the string $s$ we continue to extend $a$ with the LLM until the verifier is able to return a valid score. At this point, we can return the estimated value $v(s)$ of $s$ as follows:

$$v(s) = \texttt{Verifier}(s + a) = \begin{cases} +1 & \text{verified, but may be incomplete.} \\ -1 & \text{verified as a failure.} \end{cases} \qquad (2)$$

If $v(s) = +1$, we also add $s + a$ as a child in the tree, while if $v(s) = -1$, we do not add $s + a$ since it is a verified failure. Appendix H gives explicit examples of scoring partial programs.

**Backpropagation.**  The last step of an iteration of MCTS is to backpropagate the observed value from leaf back up to root. We do this in the standard way so that signal is propagated up the tree. The algorithm terminates when it finds a complete solution that verifies or when it exceeds some token limit or time limit.

Appendix A presents theory that VerMCTS optimizes an upper bound on the value function.

## 3 Results

A full description of the problem suite can be found in Appendix B and experimental methods in Appendix C respectively. Here we present the main results.

We run VerMCTS and our three baselines across our full suite of problems. The aggregate results are illustrated in Figure 3. In both programming languages VerMCTS convincingly outperforms the baselines. Generally, MCTS rollout is second best, followed by whole sampling and then Reflexion. As previewed in the introduction, we see about a 30% absolute improvement in pass@5000 for VerMCTS relative to whole sampling. Note that Coq is substantially more challenging since the verifier is less automated.

Examining the performance of the baselines more closely, we see that MCTS rollout does outperform whole sampling, even though the verifier is not used to guide the search at intermediate steps. But, using the verifier in VerMCTS provides even better performance. Looking at Reflexion, we see that performance is poor on these tasks. This could be due to many reasons including: (1) the base model is not good at responding to errors in low resource languages like Dafny and Coq, (2) the base model does not do well integrating the long contexts created by the Reflexion prompts, and (3) Reflexion does not make it as easy to backtrack.

Due to space constraints, extended results are in Appendix D, extended related work in Appendix E, and further discussion in Appendix F.

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

## A   Connecting the partial program score to the MDP

Importantly, while the verifier gives us ground truth information about whether the program verifies so far, it does not give an unbiased estimate of the true value of a state in the MDP defined above. Instead, we can view our use of the verifier as a heuristic that quickly returns an *upper bound* on the value function of a potential child. Recall that the value function $V^*$ of the optimal policy in a deterministic MDP with state-based rewards like ours is defined by the Bellman equation $V^*(s) = \max_a r(s) + V^*(s + a)$. With this definition, we can formally describe the optimism property of our estimates values as follows:

**Lemma A.1.** *The value $v(s)$ returned by Algorithm 1 satisfies the following:*

$$v(s) \geq \mathbb{E}_{a \sim \texttt{LLM+Verifier}|s}[V^*(s + a)] \tag{3}$$

This is fairly straightforward to prove. If $v(s) = -1$, then we know that the sampled completion $a$ is a failure no matter what happens afterwards, so $v(s) = V^*(s + a) = -1$. On the other hand, if $v(s) = 1$ then we are assigning the maximal possible value in this MDP, so $v(s) \geq V^*(s + a)$.

In this way, our value estimate is explicitly an *optimistic* estimate of the value. This is even beyond the UCT score computed by MCTS. We hypothesize that this encourages deeper exploration of the search trees which can be beneficial in the multi-step problems we consider.

## B   A problem suite for multi-step verified programming

### B.1   Defining the problems

We are not aware of any existing collections of problems that are designed for multi-step program synthesis and checked using verifiers. That is why we have created our own problem suite of nine problems. The problems represent meaningful scenarios in verified programming. They require creating Algebraic Data Types (ADTs), defining functions on them using pattern matching, and proving properties using induction. Compared to prior benchmarks, the problems require more intricate multi-step reasoning and test capabilities that are specifically important for verified programming. The problems are defined as follows:

**Factorial**  asks to define the factorial function and to prove that it is always strictly positive.

**Opt0**  asks to define an ADT for arithmetic expressions, an optimizer, and to prove that the optimizer preserves semantics.

**Opt0 Opt**  asks to define an ADT for arithmetic expressions, an optimizer, an optimal predicate, and to prove that the optimizer is optimal.

**BST**  asks to define a tree, the binary search tree (BST) property, insertion, and to prove two properties of insertions (membership and BST preservation).

**Repeat**  asks to define a function returning a list with a given element repeated a given number of times, and to prove two properties related to length and membership.

**Lights**  asks to define an ADT for traffic lights, then write a function ensuring that red and green lights are always separated by yellow lights, and then to prove its correctness.

**Food**  asks to define an ADT that represents different foods with toppings, and a predicate about the amount of toppings, and to prove a property of this predicate.

**Days**  asks to define an ADT that represents days of the week, two functions that iterate through business days, and then to prove a property of weekdays.

**Reverse**  asks to define a function that reverses a list, and prove two properties of list reversals (permutation and involution).

All problems are implemented in Dafny, and all but the last three are implemented in Coq, giving a total of 15 problems. Since the Coq verifier has substantially less automation than Dafny which leads to longer proofs and since the model is not always very consistent at Coq syntax, just for Coq we provide some syntax hints in the prompt. The full prompts can be found in Appendix G.

### B.2 Criteria for Success

In order to be considered successful, a program must first pass the verifier and some syntactic checks (e.g. the presence of a proof marker and a problem-specific minimum number of lines of code). These initial checks are meant to ensure the model has made a successful attempt to prove a lemma.

A second check ensures that the model has proven the correct lemma: In order to check whether a model has proven a property, we inject a second lemma with it, and prove it by referring to the lemma we asked the model to write. If the model has proven this lemma as directed, this new code including check lemma will verify successfully. If the model has proven an incorrect lemma, a verifier error will be produced. Note that the check lemma is only injected into the verifier input. The model does not get to see it, so this check does not provide additional hints to the model.

A full description of each problem including the prompts and lemmas used for checking success can be found in Appendix G.

## C  Experimental setup

### C.1  Pass@$T$ evaluation metric

We report all of our results in terms of pass@$T$, which is, to our knowledge, a novel metric inspired by pass@$k$ that is often used in code generation benchmarks [Chen et al., 2021]. While pass@$k$ computes the probability of generating a success when we sample $k$ programs, pass@$T$ computes the probability of success if we allow the model to sample $T$ tokens. Pass@$T$ has several benefits:

1. Pass@$T$ fairly compares methods. One run of MCTS can be much more expensive than sampling one program from a model, so using pass@$k$ is not fair. In contrast pass@$T$ really estimates the dominant cost of generation, namely how many tokens need to be generated to yield success.
2. Pass@$T$ controls for hardware and implementation variability. Compared to using wall-clock time, using pass@$T$ does not depend on the underlying hardware and system-level optimizations.

To estimate pass@$T$, we generate $n$ runs per problem of up to $T_{max}$ tokens per run (where if the run terminates successfully before $T_{max}$ we stop the run). Then for each $T \leq T_{max}$, we have $n$ binary trials indicating whether that run terminated successfully in $\leq T$ tokens. In the results, we report the mean of these $n$ binary variables and also 95% Wilson intervals [Wilson, 1927].

### C.2  Base model

VerMCTS is compatible with any base model and only requires sampling from the model (no training is needed). We opt to use an open-weights model as the base language model and then compare different sampling procedures on top of this base model. Specifically, we use Phind-CodeLLama-34B-v2 [Phind, 2023, Roziere et al., 2023]. This model has been trained explicitly for code generation, but the verified programming languages we use are relatively "low resource" languages, so the models will perform worse than at high-resource languages [Cassano et al., 2023a].

### C.3  Baselines

We consider a variety of baseline methods to illustrate the benefits of leveraging the verifier inside of VerMCTS.

- **Whole sampling.** The most naive baseline just samples entire programs from the base model. To compute pass@$T$ we just continue generating new samples until success or until the token limit is reached.
- **Rollout MCTS.** Related work on MCTS uses rollouts to evaluate a node [Chaslot et al., 2008, Zhang et al., 2023a]. We ablate the importance of using the verifier by replacing the "evaluate and maybe expand" step with separate expand and evaluate steps. We expand by sampling a fixed number of actions $k$ from the LLM and evaluate by rolling out with the LLM to a terminal node before querying the reward function.

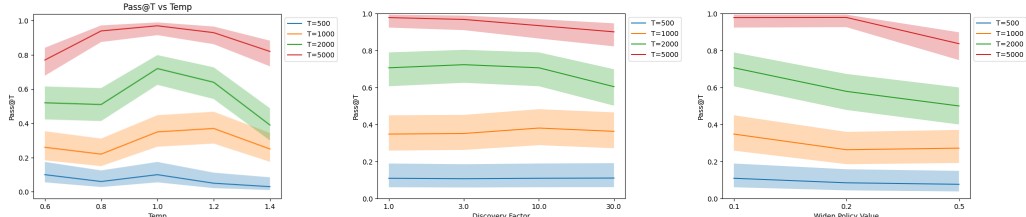

Figure 4: Hyperparameter ablations for VerMCTS on opt0 in Dafny. We find that performance is generally fairly stable to hyperparameter choices.

- **Reflexion.** Finally, to show how VerMCTS is efficient at incorporating information from the verifier we also compare to a Reflexion [Shinn et al., 2023] baseline where the LLM gets to view the errors produced by the verifier on failed attempts.

## C.4 Hyperparamters

When sampling from the LLM, we always use nucleus sampling [Holtzman et al., 2019] with $p = 0.95$ following Roziere et al. [2023]. For every method, we sweep over temperature on one representative problem and use that temperature for the rest. Our VerMCTS algorithm also introduces two hyperparameters that govern exploration: $c_{UCT}$ and $p_{widen}$ which we found fairly straightforward to set. We tune hyperparameters on one particular problem (opt0) in Dafny, but only checking for verification and not additionally checking for correctness. Each method has slightly different hyperparameters, but we generally tune temperature of the LLM, the MCTS exploration coefficient, and the MCTS prior for widen nodes. Hyperaparameters are then fixed for all other experiments. Each algorithm's parameters are described below.

**VerMCTS.** We sweep over temperature in [0.6, 0.8, 1.0, 1.0, 1.4] and find 1.0 to be best, exploration coefficient in [1, 3, 10, 30] and find 3 to be best, and the "widen policy value", i.e. the prior value of the widen nodes in [0.1, 0.2, 0.5] and find 0.1 to be best. See Figure 4.

**MCTS rollout.** We also sweep over temperature in [0.6, 0.8, 1.0, 1.0, 1.4] and find 0.8 to be best and exploration coefficient in [1, 3, 10, 30] and find 1 to be best. Note, instead of widen nodes, each node has a fixed number of children (3 in our experiments).

**Reflexion.** We sweep over temperature in [0.2, 0.4, 0.6, 0.8, 1.0] and find 0.4 to be best.

**Whole sampling.** We sweep over temperature in [0.2, 0.4, 0.6, 0.8, 1.0] and find 0.6 to be best.

We use the Transformers library Wolf et al. [2020] to query the LLMs. For the MCTS, we adapt a generic open-source library ImparaAI [2024].

# D  Extended results

## D.1  Per-problem results

In Figures Figure 5 and Figure 6, we present the per-problem results on our problem suite. There is substantial variation across problems, but across all problems VerMCTS is the best approach or within the margin of error, often exceeding the baselines by a large margin and sometimes solving problems that no baseline solves at all. That said, some problems are clearly challenging: on one problem in Dafny and three in Coq, none of the algorithms find a solution within 5000 tokens.

## D.2  Examining the VerMCTS search trees

In Figure 7 we provide an experiment to probe for a mechanistic understanding of how VerMCTS works in Dafny. We consider the number of nodes (excluding widen nodes), the depth and the width of the search trees as the number of tokens generated increases. Note that since we do not add

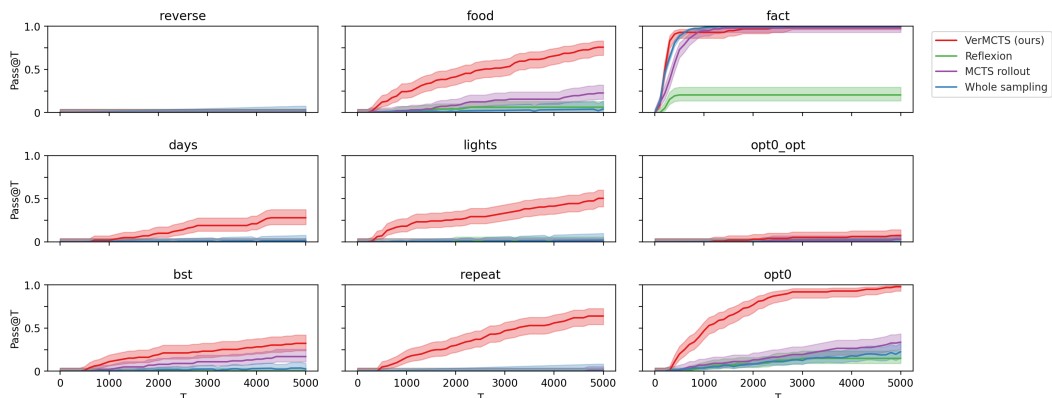

Figure 5: Pass@T results for all algorithms on our suite of problems in Dafny.

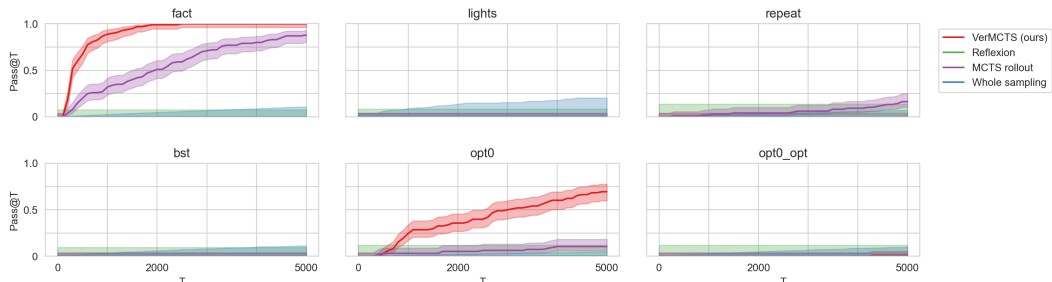

Figure 6: Pass@T results for all algorithms on our suite of problems in Coq.

failed expansions to the tree, sometimes more tokens are generated without adding nodes to the tree. Generally, we observe that the more challenging problems (with lower pass rates) tend to lead to larger search trees, indicating that the algorithm is successfull. We also notice that while the number of nodes grows fairly linearly across time for most problems, the depth grows earlier and then flattens out. This suggests that the VerMCTS search is closer to "depth first", first pushing an expansion branch to a terminal node before going back and widening the tree.

## E   Related Work

**Neural Program Synthesis with Large Language Models**   Austin et al. [2021] and Chen et al. [2021] demonstrated that Large Language Models (LLMs) can generate correct Python programs from natural language descriptions. These studies introduced the MBPP and HumanEval datasets, respectively, which are widely used for evaluating LLMs in program synthesis tasks. Cassano et al. [2023a] extended this concept by showing that LLMs can also generate programs in over 20 languages other than Python. This was achieved by translating the MBPP and HumanEval datasets using their system, MultiPL-E. Their findings indicate that generating accurate programs in lower resource languages is more challenging compared to higher resource languages, such as Python. In our experiments, for proof synthesis, we have another dimension of challenge: some languages (Coq) are inherently more challenging than others (Dafny), depending on how much automation the verifiers provide. However, none of these works explored the generation of programs that are correct by construction.

**Symbolic Algorithms for Neural Program Synthesis**   Grand et al. [2023] integrated a classic symbolic top-down synthesis algorithm for library learning Bowers et al. [2023] with LLMs. Cassano et al. [2023b] employed program decomposition and a bottom-up tree-search algorithm to infer missing TypeScript types. Zhou et al. [2023] used Monte Carlo Tree Search (MCTS) to create single-function programs in Python. Zhang et al. [2023a] applied a tree-based planning algorithm for decoding LLM token sequences, which were then evaluated for correctness using a test suite. Lample et al. [2022] adapted MCTS for neural theorem proving by employing a tree-based search

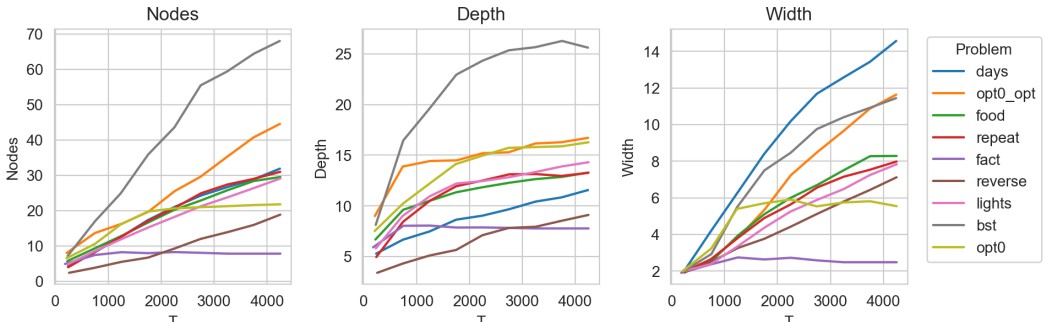

Figure 7: Average number of nodes, depth, and width of the VerMCTS search tree as the number of tokens increases across the full suite of Dafny problems. Recall that failed expansions are not added to the tree. Harder problems tend to lead to larger trees.

algorithm to generate proof trees in Lean. Different from these closely related works, we (1) focus on verified program synthesis in Dafny and Coq, and (2) leverage the verifier inside the loop of the search algorithm to efficiently guide the search.

**Theorem Proving with Large Language Models** Han et al. [2022] demonstrated that LLMs can be trained to generate proofs in Lean through self-supervision. Yang et al. [2023] presented that Retrieval-Augmented Generation (RAG) Glass et al. [2022] models significantly enhance LLMs' performance in theorem proving tasks. First et al. [2023] employed a methodology akin to that of Han et al. [2022] to generate and repair complete proofs in Isabelle/HOL. Jiang et al. [2023] introduced methods to first map natural language proofs to formal proof sketches in Isabelle and then fill in the gaps using an automated prover. These studies predominantly used LLMs to iteratively generate individual proof steps, which were then verified using a theorem prover. Thakur et al. [2023] propose a language-agent approach to formal theorem-proving, alternating selection and execution steps. In contrast, we focus on verified program synthesis and developing a method that effectively integrates a verifier and LLM without any additional training.

**Scoring Partial Programs** Desai et al. [2016], one of the first to effectively tackle the problem of program synthesis using natural language, used a scoring function to rank candidate partial programs. Cassano et al. [2023b] similarly used a scoring function to rank candidate partial programs based on their types in order to aid the tree search process, and provided multiple solutions to the user ranked by their score. Ye et al. [2021] used abstract interpretation to rule out partial programs that do not satisfy some constraints, typically on input/output examples. Chen et al. [2022] used LLM-generated unit tests suites and their pass rates to score candidate programs, and provided the user with the top-scoring program. Ni et al. [2023] further utilized execution information to rank candidate programs. Shirafuji et al. [2023] used a scoring function to rank example refactoring programs generated by an LLM before applying them to the given code. Zhang et al. [2023b] studies using scoring functions to rank candidate partial programs in-depth, and proposes the use of a *reviewer* model to score candidate programs based on how closely they match the given instruction. Most of these works have scored partial programs specified as grammatical programs with holes as opposed to our left-to-right generation of partial programs, and have not considered verified programming languages.

# F   Discussion

We have demonstrated that relatively weak language models can reliably produce verified code by guiding a search process that verifies partial programs at each step. Our technique shines on multi-step problems, made of dependent sub-problems. Our technique can be adapted to a setting where the interfaces and specifications are given, and the code is verified at each step by additional code containing assertions or proofs.

**Limitations.** A key aspect of our approach resides in the scoring of partial programs. However, the scoring is limited by coarse granularity and lack of lookahead in the scoring function. The granularity

of the verification step is a whole unit, e.g. a function in Dafny and a command in Coq. For Dafny, the coarse granularity means we have to wait multiple lines to get feedback. For Coq, the fine granularity doesn't help much with bigger proofs, which require planning.

**Future work.**  What we find most interesting and promising about our approach is that so much is possible by a "blind" search that only uses scalar reward signal. In future work, it would be fruitful to find ways of allowing the search to rely on richer feedback while maintaining the efficiency of leveraging the verifier to avoid doing costly rollouts or reflection steps. Moreover, it will be interesting to see if the basic idea of VerMCTS, using a cheap and provable upper bound on the value function to guide search, can be applied beyond the verified programming setting.

# G  Prompts

## G.1  Repeat Prompt

**Coq.** *In Coq: (1) Write a function 'repeat' that takes an integer 'x' and a natural number 'n' as inputs, and returns a list of length 'n' in which every element is 'x'. (2) Then write a lemma 'repeat_correct' that checks that for any 'x' and 'n', 'repeat' returns a list of length 'n' and that every element of the list is 'x'.*

**Dafny.** *In Dafny: (1) Write a function 'repeat' that takes an integer 'x' and a natural number 'n' as inputs, and returns a list of length 'n' in which every element is 'x'. (2) Then write a lemma 'repeat_correct' that checks that for any 'x' and 'n', 'repeat' returns a list of length 'n' and that every element of the list is 'x'.*

**Hints for Coq.**
*### Hint: Start with 'Require Import List. Import ListNotations.'*

**Check lemma for Coq.**

```
Lemma CHECK_repeat_correct: ∀ (x: int) (n: nat),
    length (repeat x n) = n /\    ∀ i, 0 ≤ i -> i < n -> nth (repeat x n) i = x.
  Proof.
    intros.
    eapply repeat_correct; eauto.
  Qed.
```

**Check lemma for Dafny.**

```
lemma CHECK_repeat_correct(x: int, n: nat)
    ensures |repeat(x, n)| = n
    ensures ∀ i • 0 ≤ i < n ⟹ repeat(x, n)[i] = x
  {
    repeat_correct(x, n);
  }
```

## G.2  Opt0 Opt Prompt

**Coq.** *In Coq, write an ADT 'Expr' for arithmetic expressions comprising constants, variables and binary addition. Then write a predicate 'optimal' that holds on an expression if it has no additions by 0. Then write an optimizer 'optimize' that removes all additions by 0. Then write a lemma 'OptimizerOptimal' that ensures 'optimal(optimize(e))' for all expressions 'e'.*

**Dafny.** *In Dafny, write an ADT 'Expr' for arithmetic expressions comprising constants, variables and binary addition. Then write a predicate 'optimal' that holds on an expression if it has no additions by 0. Then write an optimizer 'optimize' that removes all additions by 0. Then write a lemma 'OptimizerOptimal' that ensures 'optimal(optimize(e))' for all expressions 'e'.*

**Hints for Coq.**
*### Hint: In the addition case, the 'optimize' function should recursively optimize the sub-expressions and then match on the optimized sub-expressions.*
*### Hint: You can import the 'string' datatype with the line 'Require Import Coq.Strings.String.'*
*### Hint: Use Fixpoint instead of Definition for recursive functions.*
*### Hint: If you do induction on 'e' with sub-expressions 'e1' and 'e2', the two inductive hypotheses are called 'IHe1' and 'IHe2'.*

**Check lemma for Coq.**

```
lemma CHECK_OptimizerOptimal(e: Expr) ensures optimal(optimize(e)) { OptimizerOptimal(e); }
```

**Check lemma for Dafny.**

```
lemma CHECK_OptimizerOptimal(e: Expr) ensures optimal(optimize(e)) { OptimizerOptimal(e); }
```

### G.3 Lights Prompt

**Coq.** *In Coq: (1) Write a datatype 'light' for traffic lights with cases 'Red', 'Yellow', 'Green'. (2) Write a function 'activation' which takes two lights, source and target, and returns a list of lights, the first element being the source and the last element being the target. If the source and target are not yellow and are distinct, then the returned list has a middle element of yellow. (3) Write a helper 'adjacent_ok' that takes two lights, and checks that they are not one red and the other green. (4) Write a helper 'all_adjacent_ok' that takes a list of lights, and checks that all adjacent elements are 'adjacent_ok'. (5) Write a lemma 'check_activation' to prove that forall source and target lights, a returned list never has adjacent elements that are distinct and red or green. The proposition should be 'all_adjacent_ok (activation source target)'.*

**Dafny.** *In Dafny: (1) Write a datatype 'light' for traffic lights with cases 'Red', 'Yellow', 'Green'. (2) Write a function 'activation' which takes two lights, source and target, and returns a list of lights, the first element being the source and the last element being the target. If the source and target are not yellow and are distinct, then the returned list has a middle element of yellow. (3) Write a helper 'adjacent_ok' that takes two lights, and checks that they are not one red and the other green. (4) Write a helper 'all_adjacent_ok' that takes a list of lights, and checks that all adjacent elements are 'adjacent_ok'. (5) Write a lemma 'check_activation(source: light, target: light)' to prove that a returned list never has adjacent elements that are distinct and red or green. The 'ensures' clause should be 'all_adjacent_ok(activation(source, target))'.*

**Hints for Coq.**
*### Hint: Start with 'Require Import List. Import ListNotations.'*

**Check lemma for Coq.**

```
Lemma CHECK__check_activation: ∀ (source: light) (target: light),
    all_adjacent_ok(activation(source  target).
    Proof.
      intros.
      eapply check_activation; eauto.
    Qed.
```

**Check lemma for Dafny.**

```
lemma CHECK__check_activation(source: light, target: light)
    ensures all_adjacent_ok(activation(source, target))
    {
      check_activation(source, target);
    }
```

### G.4 BST Prompt

**Coq.** *In Coq, (1) write an ADT for a tree of natural numbers. Call it 'Tree'. Then (2) write a predicate 'IsBST' that checks whether a given tree is a binary search tree (BST). Then (3) write a function 'insert' that inserts an element into a binary search tree while preserving the BST property. Then (4) write a predicate 'Contains' that checks whether a given tree contains a given element. Then (5) write a lemma 'InsertContains' about the insert function that ensures that the tree resulting from inserting an element contains that element (without requiring nor ensuring the BST property). Then (6) write another lemma 'InsertPreservesBST' about the insert function that checks the BST property continues to hold after insertion. This lemma should take bounds on the BST, and require that the element to be inserted is within those bounds.*

**Dafny.** *In Dafny, (1) write an ADT for a tree of natural numbers. Call it 'Tree'. Then (2) write a predicate 'IsBST' that checks whether a given tree is a binary search tree (BST). Then (3) write a function 'insert' that inserts an element into a binary search tree while preserving the BST property. Then (4) write a predicate 'Contains' that checks whether a given tree contains a given element. Then (5) write a lemma 'InsertContains' about the insert function that ensures that the tree resulting from inserting an element contains that element (without requiring nor ensuring the BST property). Then (6) write another lemma 'InsertPreservesBST' about the insert function that checks the BST property continues to hold after insertion. This lemma should take bounds on the BST, and require that the element to be inserted is within those bounds.*

**Hints for Coq.**
*### Hint: Start with 'Require Import List. Import ListNotations.'*

*### Hint: Use Fixpoint instead of Definition for recursive functions.*
*### Hint: Use 'l' and 'r' for variable names instead of 'left' and 'right' to avoid name clashes.*

**Check lemma for Coq.**

```
// (5) Lemma about the insert function that ensures the tree resulting
//       from inserting an element contains that element
Lemma CHECK_InsertContains: ∀ (t: Tree) (x: nat),
  Contains (insert t  x) x.
Proof.
  intros.
  eapply InsertContains; eauto.
Qed.

// (6) Lemma about the insert function that checks the BST property
//      continues to hold after insertion
lemma CHECK_InsertPreservesBST: ∀ (t: Tree) (x: nat) (min: nat) (max: nat),
  (IsBST t min max) –> min ≤ x ≤ max –>
  IsBST (insert t x) min max.
Proof.
    intros.
    eapply InsertPreservesBST; eauto.
Qed.
```

**Check lemma for Dafny.**

```
// (5) Lemma about the insert function that ensures the tree resulting from
//      inserting an element contains that element
lemma CHECK_InsertContains(t: Tree, x: nat)
  ensures Contains(insert(t, x), x)
{
  InsertContains(t, x);
}

// (6) Lemma about the insert function that checks the BST property continues
//      to hold after insertion
lemma CHECK_InsertPreservesBST(t: Tree, x: nat, min: nat, max: nat)
  requires IsBST(t, min, max) ∧ min ≤ x ≤ max
  ensures IsBST(insert(t, x), min, max)
{
    InsertPreservesBST(t, x, min, max);
}
```

### G.5    Opt0 Prompt

**Coq.** *In Coq, write an ADT for arithmetic expressions (called 'Expr') comprising constants, variables and binary additions. Then write an evaluator (called 'Eval') taking an expression and an environment (a function that takes a variable name and returns a number) and returning the number resulting from evaluation. Then write an optimizer (called 'Optimize') taking an expression and returning an expression with all additions by 0 removed. Then prove that the optimizer preserves the semantics as defined by the evaluation function. Do so by proving the lemma 'OptimizePreservesSemantics: forall (e: Expr) (env: string -> nat), Eval(Optimize(e), env) = Eval(e, env)'.*

**Dafny.** *In Dafny, write an ADT for arithmetic expressions (called 'Expr') comprising constants, variables and binary additions. Then write an evaluator (called 'Eval') taking an expression and an environment (a function that takes a variable name and returns a number) and returning the number resulting from evaluation. Then write an optimizer (called 'Optimize') taking an expression and returning an expression with all additions by 0 removed. Then prove that the optimizer preserves the semantics as defined by the evaluation function. Do so by proving the lemma 'OptimizePreservesSemantics(e: Expr, env: string -> int) ensures Eval(Optimize(e), env) == Eval(e, env)'.*

**Hints for Coq.**
*### Hint: In the optimizer, recursively optimize the sub-expressions.*
*### Hint: You can import the 'string' datatype with the line 'Require Import Coq.Strings.String.'.*

*### Hint: Use Fixpoint instead of Definition for recursive functions.*
*### Hint: With tactics like 'induction' and 'destruct', _avoid_ naming with 'as' and let Coq pick the names for you. For example, use 'induction e.' but _not_ 'induction e as [...]'.*

*### Hint: For the proof, do 'induction e.'. Do NOT name the hypotheses with 'as'.*
*### Hint: The simple cases are by 'simpl. reflexivity.'.*
*### Hint: The addition case is by 'simpl. rewrite <- IHe1. rewrite <- IHe2. destruct (optimize e1); destruct (optimize e2); try destruct n; try destruct n0; eauto using PeanoNat.Nat.add_0_r.'.*
*### Hint: You'll need 'Require Import Arith'.*

**Check lemma for Coq.**

```
Lemma CHECK_OPS: ∀ (e: Expr) (env: string -> nat), Eval (Optimize e) env = Eval e env.
    Proof.
    intros.
    apply OptimizePreservesSemantics; eauto.
    Qed.
```

**Check lemma for Dafny.**

```
lemma CHECK_OPS(e: Expr, env: string -> int)
    requires true
    ensures Eval(Optimize(e), env) = Eval(e, env)
{
    OptimizePreservesSemantics(e, env);
}
```

## G.6   Factorial Prompt

**Coq.** *In Coq, write a factorial function, called 'fac', and prove (in a lemma 'FacPositive: forall (n: nat), fac n > 0') that the factorial is always strictly positive.*

**Dafny.** *In Dafny, write a factorial function, called 'fac', and prove (in a lemma called 'FacPositive(n: nat)') that the factorial is always strictly positive.*

**Hints for Coq.**
*### Hint: Don't forget to import the Arith module.*
*### Hint: use 'Nat.lt_0_1' in the base case of the proof.*
*### Hint: use 'Nat.lt_lt_add_r' in the inductive case of the proof.*

**Check lemma for Coq.**

```
Lemma CHECK_FacPositive: ∀ (n: nat), fac n > 0. Proof. intros. apply FacPositive; eauto. Qed.
```

**Check lemma for Dafny.**

```
lemma CHECK_FacPositive(n: nat) ensures fac(n) > 0 { FacPositive(n); }
```

## G.7   Food Prompt

*In Dafny: (1) Write a datatype for 'Food': 'Pasta' or 'Pizza'. Each Pasta or Pizza has a list of toppings. Each 'Topping' is one of: 'tomato', 'cheese', 'olive', 'broccoli', 'mushroom', 'pepper'. (2) Write a predicate 'ok' that accepts any pizza with five toppings or fewer, and any pasta with two toppings or fewer. (3) Write a lemma 'ok3_pizza' that proves that an accepted food with three or more toppings must be a pizza.*

**Hints for Dafny.**
*### Hint: The length of a list or sequence 's' is '|s|'.*

**Check lemma for Dafny.**

```
lemma CHECK_ok3_pizza(x: Food)
    requires ok(x)
    requires |x.toppings| ≥ 3
    ensures match x { case Pizza(_) ⇒ true case _ ⇒ false }
    {
      ok3_pizza(x);
    }
```

## G.8   Reverse Prompt

*In Dafny: (1) Write a function 'reverse' that takes a list as input and reverses it. (2) Then write a lemma 'reverse_permutes' that checks that for any list 'l', an element exists in 'l' if and only if it exists in the result of calling 'reverse' on 'l'. (3) Then write a lemma 'reverse_involutes' that checks that for any list 'l', calling 'reverse' twice on 'l' yields 'l'.*

**Hints for Dafny.**
*### Hint: The length of a list or sequence 's' is '|s|'.*
*### Hint: Use a plain 'function' to define 'reverse', not a 'function method' or a 'method'.*

**Check lemma for Dafny.**

```
lemma CHECK__reverse_permutes(l: seq<int>)
    // TODO
    {
    }
    lemma CHECK__reverse_involutes(l: seq<int>)
    ensures reverse(reverse(l)) = l;
    {
      reverse_involutes(l);
    }
```

## G.9   Days Prompt

*In Dafny: (1) Write an ADT 'Day' for the days of the week: 'Sunday' to 'Saturday'. (2) Write a function 'next_biz_day' that gives the next business day. (3) Write a function 'iter_biz_day(d: Day, n: nat): Day' that iterates the next business day function, for an arbitrary number n of business days. (4) Write a lemma 'iter5_biz_day_idempotent' that ensures that starting with a business day, taking the next five business days is idempotent.*

**Check lemma for Dafny.**

```
lemma CHECK_iter5_biz_day_idempotent(d: Day)
    requires d ≠ Saturday
    requires d ≠ Sunday
    ensures iter_biz_day(d, 5) = d
    {
      iter5_biz_day_idempotent(d);
    }
```

# H   Examples of Scoring Partial Programs

Partial program with a score of $0$:

```
datatype Expr =
```

Partial program with a score of $+1$:

```
datatype Expr =
    | Const(val: int)
```

Partial program with a score of $-1$:

```
datatype Expr =
    | Const(val: int)
    | Var(name: string)
    | Add(e1: Expr, e2: Expr)

function Evaluate(e: Expr,
    env: string -> int): int
    reads env
{

    match e
    case Const(val) ⇒ val
    case Var(name) ⇒ env(name)
    case Add(e1, e2) ⇒
      Evaluate(e1, env) +
      Evaluate(e2, env)
}
```

The negative score is due to the `reads` clause, which shouldn't be there. Unfortunately, we only confirm the error once the whole function is generated.

# I  Broader Impacts

The development of algorithms that allow generation of verified code using smaller models has notable broader impacts on both machine learning and society. We increase the efficiency per token in code language model usage, and allow for the usage of smaller models. This further reduces energy consumption and allows for a usage of cheaper hardware, thereby democratizing access to this technology. Our approach, which is asking models to prove their work is correct, and then immediately and externally checking whether the proof is correct, can mitigate some of the open issues with trusting LLMs.

