# OpenReview forum: "VerMCTS: Synthesizing Multi-Step Programs using a Verifier, a Large Language Model, and Tree Search"
_NeurIPS.cc/2024/Workshop/MATH-AI — MATH-AI 24_

### Official Review · Reviewer_DQGp · 2024-09-29

**Rating:** 8
**Confidence:** 4

**Review:**

### Summary

This work tackles the problem of automatic verified programming by augmenting a Monte Carlo Tree Search process with expansions driven by an LLM and evaluation carried by logical verifiers (Dafny and Coq). In particular, the sound and deterministic logical verifiers help place an upper bound on the value function for partial programs during the search, and also help avoid repeating futile partial programs. The presented approach also leverages progressive widening to balance exploitation and exploration, as well as to reduce costly LLM calls. The approach is shown to be effective and to outperform alternatives.

### Strengths

The authors did a good job conveying the motivation, idea, execution, and evaluation of VerMCTS. I appreciate the setup of the problem of synthesizing not only proofs but also programs and specifications, and I like the use of an additional lemma to mechanically guarantee the quality of the synthesized specification. The components of the approach are reasonable and fit well together. Particularly, I find modifying MCTS to first evaluate and then possibly expand intriguing.

### Weaknesses

I would appreciate it if the language of the paper can be improved by eliminating typos (listed below), and if the authors could further shed light on how they might address the limitations identified (which I appreciate) and my remarks below.

### Remarks

- At the evaluate and maybe expand step, the LLM needs to be invoked until the verifier can give a definite score. Say, in the case of Coq, I wonder if it could ever be the case that the LLM keeps generating valid tactics that sometimes do not make meaningful proof progress (e.g., `repeat`, `destruct`, `idtac`), thus wasting time and the allotted number of tokens. If so, how much do the authors think this played a role in the efficacy of their approach, and would they have any countermeasures? It would be even better if the authors could evaluate on this.

- The authors have identified a limitation to be Dafny and Coq being low-resource programming languages. While the point of this work is to not require additional training, do the authors have plans to address that limitation? I would be curious to see how much the results can be improved if the authors were to adopt a more fine-tuned model.

- Could the authors elaborate on why they decided to target Dafny and Coq? Those are two pretty different tools and, as the authors pointed out, have different levels of automation. If one were to maximally optimize a technique like the one the authors are proposing, I would ideally like to see more justification for the target domain and work done or will be done to optimize.

### Typos

- Fig. 1 (c): "Once we have a value ~~and maybe~~ from ..."
- Line 36: Redundant "introduce a suite of"
- Fig. 2: Missing comma after "Add to tree" in the diagram
- Line 122: Missing period after "We will use the verifier"?
- Line 291: "The problems represent ~~a~~ meaningful scenarios in verified programming."
- Line 320: "... has made a successful attempt _to_ prove a lemma."
- Line 350: "This model_s_ has been ..."
- Fig. 4: "~~Perfromance~~ Performance"
- Line 400: "... indicating that the algorithm is succesful~~ly~~."

---

### Official Review · Reviewer_YUuY · 2024-10-03

**Rating:** 4
**Confidence:** 4

**Review:**

# Summary

The paper introduces VerMCTS, which combines LLM and an MCTS variant to synthesize programs (and corresponding proofs) in verification-native languages (Dafny and Coq), and proposes a test set consisting of 15 problems (9 in Dafny, 6 in Coq). On the test set, VerMCTS performs better than reflexion, sampling, and MCTS rollout given a fixed budget of tokens up to 5k.


# Strengths


## Quality

1. The design of the MCTS variant described in Section 2.2 is well-conceived. The introduction of the "widen" node and the UCT prior effectively balances deeper search exploration while retaining the flexibility to explore in breadth, enhancing the overall search efficiency.
2. The performance improvement of VerMCTS on the proposed test set is significant.


# Weaknesses

## Originality

VerMCTS is in fact tackling 2 tasks together: program synthesis and proof synthesis. While in today's neural program synthesis, search is less employed (for good reasons, which I'll elaborate below in the Quality Section), the proof synthesis community has been extensively studying various combinations of neural proof step generators and search algorithms [1]. An early attempt at Coq was made in 2017 [2], and MCTS variants have been introduced in 2022 [3]. Given the background, VerMCTS does not introduce substantial novelty regarding proof synthesis.

[[1] A Survey on Deep Learning for Theorem Proving](https://arxiv.org/abs/2404.09939)

[[2] Learning to Prove Theorems via Interacting with Proof Assistants](https://arxiv.org/abs/1905.09381)

[[3] HyperTree Proof Search for Neural Theorem Proving](https://arxiv.org/abs/2205.11491)



## Quality

1. Following up on the concerns raised in the Originality Section, the VerMCTS's contribution to the program synthesis stage is unclear. Take this task in the test set: **"In Coq, write a factorial function, called ‘fac‘, and prove (in a lemma ‘FacPositive: forall (n:652 nat), fac n > 0‘) that the factorial is always strictly positive."** for example, Coq can't reason about the function `fac` if it's only partially defined (e.g. some cases are defined but others missing). Consequently, VerMCTS would only receive feedback after it samples a complete `fac` function and starts to write the proof `FacPositive`.

    (a). This is precisely the reason why search is less employed in today's neural program synthesis — there are no good algorithmic ways to evaluate arbitrary partial programs. Although LLM judges or specialized reward models might eventually fulfill this role, an algorithmic verifier is not suitable for such tasks.

    (b). Moreover, it becomes difficult to distinguish whether a failure occurs during the program synthesis phase or the proof synthesis phase, as mistakes in either could result in verification failure. The paper would benefit from careful ablation studies, evaluating the effectiveness of VerMCTS in the program synthesis and the proof synthesis stages separately.


2. The experiments demonstrating the effectiveness of VerMCTS are conducted solely on Phind-CodeLLama-34B-v2, without ablation studies on the size or capability of the base model. In particular:

    (a). To what extent can VerMCTS improve the performance when used with a larger and more capable base model? In my own test with ChatGPT-4o on the web interface, it can already solve **7/9** Dafny problems (failed at `Opt0 Opt` and `Reverse`) at the first attempt disregarding minor syntax discrepancies (e.g. using `∧` for `&&`, using `function method` for `function`). Given the same amount of computation (not tokens), will VerMCTS still outperform reflexion and sampling when the base model is GPT-4o?

    (b). To what extent can VerMCTS improve the performance when used with a small model and we can afford more search? In proof synthesis, small models are often used in combination with a wider beam search. For example [4] used a find-tuned ByT5-299M with a beam width of 64 and [5] used Pythia 2.8B with a beam width of 32. Given the same amount of computation (not tokens), will VerMCTS outperform simple beam search or MCTS rollout when the base model is less than 1B?



[[4] LeanDojo: Theorem Proving with Retrieval-Augmented Language Models](https://arxiv.org/abs/2306.15626)

[[5] LLMSTEP: LLM proofstep suggestions in Lean](https://arxiv.org/abs/2310.18457)

3. As highlighted in the 2nd bullet point, ChatGPT-4o was able to solve **7/9** problems in Dafny at the first attempt. This small and relatively simple test set doesn't offer substantial evidence that VerMCTS's performance gain can be generalized to more diverse and complex problems. To provide a more robust evaluation, I recommend testing VerMCTS on a larger and more challenging benchmark, such as DafnyBench [6].

[[6] DafnyBench: A Benchmark for Formal Software Verification](https://arxiv.org/abs/2406.08467)


## Clarity

It would be beneficial to include an example trace illustrating how VerMCTS solves a particular problem (e.g. the motivating example). This would help clarify the approach, especially the feedback and state evaluation mechanics.



## Significance

Considering the limited novelty of VerMCTS in the proof synthesis stage, the concerns regarding its effectiveness in the program synthesis stage, the absence of ablations on the model size/ability, and a small and relatively simple test set, the paper offers little insight into advancing methodologies for either program or proof synthesis.



# Questions

1. Could you elaborate on the decision to address program synthesis and proof synthesis together, without decoupling their evaluation?

2. Building on the 2nd bullet point of the Quality Section, I would be interested in your further thoughts on the trade-offs between model size and search width. Given the same amount of computation, how would you compare the performance of a large model with fewer searches, a small model with extensive searches, and a medium-sized model with a balanced search strategy? How might the comparison vary across different kinds of problems or complexity levels?

---

### Official Review · Reviewer_MBm5 · 2024-10-06

**Rating:** 5
**Confidence:** 3

**Review:**

The paper proposes VerMCTS, which uses a logical verifier (Dafny and Coq) to guide a modified MCTS for code generation.

Pros:
 - The paper develop a new evaluation suite of 15 multi-step verified programming problems in Dafny and Coq.
 - VerMCTS leads to over 30% performance improvement in pass@5000tokens compared to repeated sampling baselines.

Cons:
 - The idea of combining MCTS and language model generation is not new. Using verifier to prune wrong leaf nodes is also straightforward.
 - The constructed evaluation suite only contains 15 problems and it seems the diversity is limited.

---

### Official Review · Reviewer_peBX · 2024-10-07

**Rating:** 7
**Confidence:** 4

**Review:**

This paper proposes VerMCTS, a modified Monte Carlo Tree Search that uses LLMs to generate verified programs using a formal verifier as the reward. VerMCTS uses progressive widening to deal with the huge action space defined on a unit of a program by giving the search a chance to widen its branches. VerMCTS uses formal verifiers to verify partial programs to avoid expensive rollouts. Experiments on 9 selected problems show that VerMCTS consistently outperforms MCTS with full rollouts and whole sampling in terms of pass@T.

**Strengths:**

Formally ensuring the soundness of LLM-generated programs is an important topic to study and there have been few papers addressing it.
The improvements from VerMCTS are very significant for the problems evaluated.

**Weaknesses:**

W1. The number of problems evaluated is very small and the problems may be prone to contamination, which undermines the validity of the argument.
Only 9 problems are used for evaluating VerMCTS, which is much fewer compared to other studies on the same topic such as the 60 problems in CloverBench [1] and the 100+ problems in [2].

W2. Might be better to have more verifier-assisted baselines.
For example, whole sampling until the verifier returns -1 or depth-first search with the verifier.


[1] Clover: Closed-Loop Verifiable Code Generation, https://arxiv.org/abs/2310.17807

[2] Towards AI-Assisted Synthesis of Verified Dafny Methods, https://arxiv.org/abs/2402.00247

---

### Decision · Program_Chairs · 2024-10-09

Accept